# Evidence for the Effectiveness of Psychological Interventions for Internalized Stigma among Adults with Schizophrenia Spectrum Disorders: A Systematic Review and Meta-Analyses

**DOI:** 10.3390/ijerph20085570

**Published:** 2023-04-18

**Authors:** Shankar Jagan, Tuti Iryani Mohd Daud, Lip Choy Chia, Suriati Mohamed Saini, Marhani Midin, Ng Eng-Teng, Selvasingam Ratnasingam

**Affiliations:** 1Department of Psychiatry, Faculty of Medicine, Universiti Kebangsaan Malaysia Medical Centre, Kuala Lumpur 56000, Malaysia; 2Hospital Canselor Tuanku Muhriz, Jalan Yaacob Latif, Bandar Tun Razak, Kuala Lumpur 56000, Malaysia; 3Department of Psychiatry & Mental Health, Sarawak General Hospital, Ministry of Health Malaysia, Sarawak 93586, Malaysia; 4Department of Psychiatry and Mental Health, Hospital Keningau, Peti Surat 11 Jalan Apin-Apin, Keningau 89007, Malaysia

**Keywords:** internalized stigma, internalised stigma, psychological interventions, therapies, schizophrenia spectrum disorders, NECT, systematic review, meta-analyses

## Abstract

In recent years, psychological interventions have been used to alleviate internalized stigma in people with schizophrenia spectrum disorders, but outcomes have been inconsistent. The aim of this review was to examine the existing evidence regarding this matter. Four electronic databases (EMBASE, MEDLINE, PsycINFO, and the Cochrane Central Register of Controlled Trials) were searched from inception until 8 September 2022, using appropriate strategies. The eligibility, quality, and strength of evidence of each study were all evaluated against the predetermined standards. Further quantitative analyses were performed using the RevMan software. A total of 27 studies were included in the systematic review. Eighteen studies with extractable data for meta-analysis yielded a statistically significant overall effect (Z = 3.00; *p* = 0.003; 95% CI: −0.69 [−1.15, −0.24]; n = 1633), although there was considerable heterogeneity (Tau^2^ = 0.89; Chi^2^ = 303.62, df = 17; *p* < 0.00001; I^2^ = 94%). Subgroup analyses for Narrative Enhancement and Cognitive Therapy (NECT) produced a statistically significant and highly homogenous effect (Z = 3.40; *p* = 0.0007; 95% CI: −0.44 [−0.70, −0.19]; n = 241; Tau^2^ = 0.00; Chi^2^ = 0.14, df = 2 (*p* = 0.93); I^2^ = 0%). In conclusion, the majority of the psychological interventions are successful in lowering levels of internalized stigma, especially NECT, and interventions that integrate multiple therapies may be more beneficial.

## 1. Introduction

Internalized stigma occurs when an individual with a mental health condition adopts stigmatizing attitudes about their condition as their own. This acceptance can have a negative impact on the person’s sense of self-worth and ability to recover from their illness [1,2,3]. Included in the category of internalized stigma are both felt/perceived stigma and self-stigma [4]. This is consistent with the regressive model of self-stigma proposed by Corrigan, which comprises four stages [5]. First, one becomes aware of the stigma of mental illness (also called “perceived stigma”), then one agrees with the stigma, one subsequently applies the stigma to oneself, and finally one experiences the negative effects of the stigma on one’s self-esteem and self-efficacy, which leads to shame (also known as “self-stigma”). Thus, according to this model, perceived stigma is the first stage of self-stigma formation. Both perceived stigma and self-stigma are involved in the internalization of public stigma. Thus, both perceived stigma and self-stigma are classified as internalized stigma. Therefore, it corresponds with the categorization proposed by Livingston and Boyd. Internalized stigma has great inhibitory effects on the intention of seeking help [6,7].

Schizophrenia is a renowned, serious mental illness that affects a large number of people in society. According to the WHO in early 2022, schizophrenia affected approximately 24 million people, or 1 in 300 people (0.32%) worldwide [8]. Males and females are equally affected; however, the peak ages of onset differ: males are affected between the ages of 10 and 25 years, while females are affected between the ages of 25 and 35 years [9]. Suicide accounts for 4 to 10% of all deaths among those diagnosed with schizophrenia, predominantly affecting young males [10]. This could be due to a variety of factors, including the reaction to the psychosis and the experience of stigma.

The World Health Organization has identified three main points regarding stigma and mental illness: (1) stigma is the primary reason for discrimination against and rejection of people with mental illness; (2) stigma has negative effects on both the prevention of mental health problems and the treatment and care of those who suffer from them; and (3) stigma violates human rights [11]. According to several studies, people with schizophrenia are more likely to experience and suffer from self-stigma than people with other mental illnesses [12,13]. Forty-one percent of a sizable European sample of adults with a schizophrenia-spectrum diagnosis reported high levels of internalized stigma, while 69 percent reported moderate or severe perceived discrimination [14]. A systematic review conducted on persons with schizophrenia spectrum disorders found that approximately 65% of participants felt stigmatized, and 56% had experienced it [15]. Stigmatization of those diagnosed with schizophrenia is widespread, and it may stem from skewed assumptions regarding their physical appearance and inconsistent acts that are associated with altered thoughts, perceptions, and behaviors [16]. Individuals having schizophrenia may experience exclusion by members of society and misconduct displayed against them regarding their emotions, thoughts, and behaviors [17]. Debilitating symptoms associated with the condition are labelled as “madness” by society, which leads to stigmatization, discrimination, and a decrease in productivity, all of which have a significant negative impact on the patient’s emotional, social, and economic well-being [18]. The majority of people diagnosed with schizophrenia reported having a low level of perceived social support, with support from significant others ranking the lowest, followed by support from friends and family [19]. Even when functional remission is reached, there is still a high unemployment rate among people with schizophrenia [20]. Prior research demonstrated that people with schizophrenia are disproportionately affected by internalized stigma [21,22]. The impaired functioning that is common in schizophrenia presents significant difficulties for both the affected person’s family and the surrounding society. Hence, among the vital goals of treatment for people with schizophrenia is to enhance their capacity to function and become self-sufficient through enhanced social adaptation [23,24].

Psychological interventions aiming to reduce the detrimental effects of internalized stigma on individuals with schizophrenia spectrum disorders have garnered considerable attention in recent years. The results of several systematic reviews and meta-analyses published in the last few years on interventions to diminish internalized stigma are still inconsistent [25,26,27,28]. This might be due to a lack of prior experimental trials that have been conducted to establish the efficacy of a certain intervention. Consequently, it is impossible to reliably verify the pooled effect sizes. In the early stages of an intervention, the lack of research from various locations around the world may reduce our understanding of its potential overall efficacy. At present, various psychological interventions have been developed and tested to reduce internalized stigma in patients with schizophrenia spectrum disorders, such as Group Psychoeducation [29,30,31,32] and Narrative Enhancement and Cognitive Therapy (NECT) [33,34]. Such treatments are crucial because they can target a wide variety of factors that relate to the psychological process of the production of internalized stigma, which was mentioned previously. The outcomes of interventions at different time intervals or in different regions varied greatly. This might be because of numerous variables such as small sample sizes, variation in intervention implementation, and anecdotal definitions of stigma. Considering this discrepancy, further study on this topic is still necessary to learn more about the therapies that have been developed so far, and including newer studies may lead to a more definitive conclusion.

### Aims

The aim of this study is to perform an updated systematic review and meta-analysis of the evidence for the effectiveness of interventions for internalized stigma among adults with schizophrenia spectrum disorders and to determine the efficacy of interventions on internalized stigma reduction among adults with schizophrenia spectrum disorders.

## 2. Materials and Methods

The systematic review and meta-analysis have been carried out in compliance with the Preferred Reporting Items for Systematic Reviews and Meta-Analyses (PRISMA) guideline [35,36]. It was registered in the International Prospective Register of Systematic Reviews (PROSPERO) on 26 August 2021, with registration number CRD42018106359 [37].

### 2.1. Search Strategy

The search was conducted across four electronic databases: EMBASE, MEDLINE, PsycINFO, and the Cochrane Central Register of Controlled Trials. Materials that existed from inception until 8 September 2022 were obtained from the selected electronic databases by using their own designed search strategy made from phrases with acceptable truncation (indicated with asterisks) and Boolean operators of AND and OR. The related search terms were “adults”, “Not adolescents, elderly”, “schizophrenia”, “psychosis”, “delusional”, “schizoaffective”, “mental illness”, “mental disorder”, “perceive stigma*”, “perceived stigma*”, “selfstigma”, “self stigma”, “internalized stigma”, “internalised stigma”, “shame”, “stigma intervention*”, “stigma reduction”, “antistigma”, “anti-stigma”, “destigmatisation”, “destigmatization”, “educat*”, “attitude change”, “NECT”, “CBT”, “photovoice”, “coming out proud”, “self disclosure”, “psychoeducat*”, “depression literacy”, “pamphlet*”, “brochure*”, “booklet”, “contact-base*”, “video-base contact”, “video-based contact”. The search method was restricted to English-language publications and research involving human participants. No geographical limitations were imposed. A bibliographic search from relevant systematic reviews and acceptable studies was carried out at the start of the research to maximize the likelihood of finding relevant studies.

### 2.2. Eligibility Criteria

Studies included in the review met the following criteria: (1) used a randomized controlled trial or similar type study design (pseudorandomized controlled trial and non-randomized experimental trial); (2) includes individuals ranging in age from 18 to 65 years old; (3) evaluated interventions for reducing internalized stigma; (4) quantitatively assessed the reduction in internalized stigma; and (5) published full papers in the English language. The exclusion criteria were (1) non-English language papers, (2) case reports, reviews, unpublished research, conference abstracts, trial protocols, and proceedings, and (3) studies that did not include people from the diagnoses of schizophrenia spectrum and other psychotic disorders.

### 2.3. Selection Process

All of the collected search results were subsequently transferred to reference management software [Covidence systematic review software, Veritas Health Innovation, Melbourne, Australia. www.covidence.org (accessed on 8 August 2022)]. This program will automatically sort out any duplicate studies upon import, making it much simpler for two independent reviewers to examine the titles and abstracts of the research. Thereafter, the eligibility of the entire texts was meticulously examined based on the inclusion and exclusion criteria that had been established previously. When there were discrepancies between the two reviewers (SJ and other members of the team), a third reviewer or the group’s consensus was used to reach a decision.

### 2.4. Methodological Quality

The McMaster Critical Appraisal Tool for Quantitative Studies was utilized in order to conduct an evaluation on the level of quality present in each of the studies that were included [38]. It consists of 15 questions that evaluate the subject areas of research purpose, literature, design, samples, outcomes, interventions, results, conclusions, and therapeutic implications. Each question was given a grade of “Yes,” “No,” or “Not Addressed,” and a score of one point was awarded for “Yes” responses. A maximum score of 15 is possible. The Australian National Health and Medical Research Council (NHMRC) evidence hierarchy was utilized to ascertain the level of evidence possessed by each study that was incorporated into the review [39]. The five aspects that were evaluated were (i) evidence base; (ii) consistency of findings across included studies; (iii) clinical impact; (iv) generalizability; and (v) applicability. In order to assist in providing direction for the overall weighting of the suggestion, each component was assigned a grade ranging from “A” to “D.”

### 2.5. Data Extraction and Management

The data extraction was carried out by two independent reviewers (SJ and TI) using spreadsheets created in Microsoft Excel (Microsoft Corporation, Redmond, WA, USA). The descriptive data that were extracted from the included studies include the following: country of origin; study setting; sample size and characteristics (diagnoses, gender, age group); exposure, comparator, and characteristics of interventions (type, frequency, duration); outcome measures and its relevant results. If further information was required, the respective authors were contacted. Any discrepancies that were found in the data that was extracted were resolved by a third reviewer (SS).

### 2.6. Statistical Analysis

A meta-analysis was carried out in order to compare the interventions that have been utilized with the purpose of lowering levels of internalized stigma. The RevMan software program (Review Manager (RevMan). Version 5.4, The Cochrane Collaboration, 2020) was used to analyze the data. For continuous outcomes measured on a variety of scales, the pooled effect was computed as a standardized mean difference (SMD). The confidence intervals (CI) were set at 95%. The I^2^ index statistic was utilized in order to provide an expression of the heterogeneity that existed among the studies. A value of 0% for the I^2^ index showed that there was no observable heterogeneity; a value of 25% indicated low heterogeneity, 50% indicated moderate heterogeneity, and 75% indicated high heterogeneity [40]. Because of the diversity of the studies that were incorporated into the analysis, a random-effect model was used so that any potential heterogeneity could be accounted for. A *p*-value of ≤0.05 was used to indicate statistical significance. Moreover, publication bias was investigated using a funnel plot.

## 3. Results

### 3.1. Systematic Review

#### 3.1.1. Study Selection

Searches conducted using online databases yielded a total of 681 studies, of which 21 were duplicates. There was a total of 276 studies retrieved from EMBASE, 79 studies from MEDLINE, 105 studies from PsycINFO, 105 studies from the Cochrane Library, and 116 studies sourced from additional records discovered through the reference list. Following the removal of studies that were found to be duplicates and a screening of the titles and abstracts, a total of 638 studies were excluded. A total of 43 studies with full-text manuscripts were evaluated for eligibility. The overall selection process is illustrated in Figure 1. Finally, a total of 27 studies were included in the systematic review.

#### 3.1.2. Quality Assessment

The level of evidence provided by the included studies was evaluated with the use of the NHMRC evidence hierarchy. Eighteen studies were rated level II (a randomized controlled trial) [29,30,32,33,34,41,42,43,44,45,46,47,48,49,50,51,52,53], while three were III-1 (a pseudorandomized controlled trial) [41,42,43]; and six were III-2 (a comparative study with concurrent controls—non-randomized, experimental trial) [31,44,45,46,47,48]. The McMaster Critical Appraisal Tool for Quantitative Studies was used to evaluate the risk of bias, and the results were expressed in percentage based on the number of criteria that were met (refer to Appendix A). Nine studies were rated >90% [32,33,41,49,50,51,52,53,54], sixteen studies were rated 80–90% [30,31,34,47,48,49,50,51,52,55,56,57,58,59,60,61], while two studies were rated 50–60% [29,55]. The manner in which randomization was conducted was described in each of the included RCTs. All of the studies also stated their findings in terms of statistical significance, and the analysis methods were appropriate. Avoidance of contamination (item 9) and avoidance of cointervention (item 10) were not reported in most of the included studies.

#### 3.1.3. Study Characteristics

Study characteristics are summarized in Table 1. Studies from fifteen different countries, including the United States of America, Canada, China, Vietnam, Japan, Taiwan, Turkey, Jordan, Israel, England, Sweden, Croatia, Finland, Spain, and Germany, were included in this review. Twenty-two studies (81.5%) were published in the last 10 years (2013–2022), with nine of those studies published since the year 2020. This review included a total of 2975 participants, the majority of whom fell into the age range of 30 to 50 years old; the number of male participants was slightly higher compared to female participants, as well as the vast majority of the participants possessed at least secondary education.

There were 15 studies composed of individuals who were treated as outpatients [29,31,33,41,43,45,47,48,49,50,52,54,57,58,60]. There were six studies conducted on participants who were inpatients [30,32,42,46,53,55]. Four studies encompassed participants from both inpatient and outpatient settings [34,43,54,56], while two studies did not address this [48,52].

There were 10 studies that only included people who had the diagnosis of schizophrenia in their participant pool [29,32,41,44,46,47,49,50,55,57]. Three studies had samples that have either schizophrenia or schizoaffective disorder as their diagnosis [30,31,54]. Nine studies had the majority of patients diagnosed with a condition that falls within the schizophrenia spectrum disorder [33,43,51,52,53,59,60,61,62]. Two studies used samples that met the criteria for serious mental illness (SMI); however, exact diagnosis was not stated but claimed that the majority of the participants had a psychotic condition [34,48]. A total of three studies had a larger number of samples consisting of individuals with affective disorders [42,45,56].

#### 3.1.4. Characteristics of Interventions Used

Group Psychoeducation was by far the most common type of intervention employed, with a total of five studies. [29,30,31,32,41]. Narrative Enhancement and Cognitive Therapy (NECT) was the second most common type of intervention utilized (total of four studies) [33,34,48,54]. Two studies reported on the Ending Self-Stigma (ESS) psychoeducational intervention [51,61]. Another two studies utilized the Self-stigma Reduction Program [57,62].

In terms of the mode of delivery, there were a total of 20 studies that involved sessions that were carried out in groups [29,30,31,32,33,34,41,42,43,46,49,50,51,54,55,57,58,59,60,61], six studies involved sessions conducted individually [43,50,53,55,59,62], and only one study involved a combination of both [57].

As for the controls utilized, a total of 19 studies compared the interventions with Treatment as Usual or a Waiting List, which consisted of patients receiving their standard psychiatric or psychosocial treatments [30,31,32,33,34,41,42,43,48,50,52,53,54,56,57,58,59,60,61]. However, one of the studies also acknowledged that their Treatment as Usual was minimally enhanced by a brochure about internalized stigma [51]. The following were among the controls used: Health and Wellness Intervention [61], Newspaper Reading [57], Group Psychoeducation [56], Face-to-face Interview [50], Supportive Sessions [29], Individual Psychoeducation [53], and Supportive Group Therapy [54]. One study used an Active Control Group which consisted of a No-Music Educational Mental Illness Stigma Talk-based Session, and Wait-list Control Group which consisted of rock and roll bingo [42].

#### 3.1.5. Characteristics of Outcome Measures Used

Table 1 provides the list of outcome measurements that were utilized in each study. There were 20 studies that used The Internalized Stigma of Mental Illness (ISMI) Scale to examine the changes in the participants’ internalized stigma [32,33,41,42,43,44,45,46,48,49,50,51,52,54,56,57,58,59,60,61]. In general, this scale possesses good construct validity, content validity, internal consistency, and test–retest reliability. However, it is necessary to take into consideration the fact that several versions and translations were utilized across these studies.

Among the other main scales used to measure the efficacy of the interventions were the following: Perceived Stigma Questionnaire [30]; the Chinese Self-stigma of Mental Illness Scale [57]; the Self-Stigma of Mental Illness Scale-Short Form and the Rosenberg Self-Esteem Scale [34]; the Stigma Towards Schizophrenia scale [55]; the Link’s Stigma-Devaluation Scale [29]; the Stigma Scale [42]; and the Japanese version of the Social Distance Scale [31]. In each of these studies, the validity and reliability of these measures were acknowledged as being sound.

#### 3.1.6. Results of the Interventions

The findings of the interventions for internalized stigma are summarized in Table 2. Overall, 15 out of 27 studies presented statistically significant outcomes. Four out of the five studies found that Group Psychoeducation was effective in achieving a statistically significant reduction in internalized stigma. One study compared Group Psychoeducation with Supportive Session, and the results favored Group Psychoeducation [29]. Another three studies that compared Group Psychoeducation with Treatment as Usual or a Waiting List produced significant reductions in internalized stigma as well [31,32,41]. However, one study that compared this intervention with Treatment as Usual provided results of lowering stigma in both groups, but considerably more so in the control group, which appeared to be a negative treatment effect for this intervention [30].

In the case of NECT, two out of the four studies revealed findings that were favorable to this intervention in comparison to Treatment as Usual or the Waiting List [34,48]. One study showed that NECT was not more effective than Treatment as Usual [33], while another study that compared this intervention with Supportive Group Therapy yielded results that favored the intervention; however, it was not statistically significant [54].

One study that compared ESS to Treatment as Usual generated data that showed that ESS was beneficial in helping to lessen essential features of internalized stigma; however, it was not statistically significant [51]. In another study, the findings indicated that ESS was not more beneficial than the Health and Wellness Intervention [61].

One study that compared the Self-stigma Reduction Program to Newspaper Reading found results that favored the intervention, although the difference was not statistically significant [57], while the findings of another study that compared this intervention to Treatment as Usual provided statistically significant results that favored this intervention [62].

Other studies that resulted in significant findings involve the following types of interventions: Antistigma Photovoice Program [52]; Family Schizophrenia Psychoeducation Program [55]; Mindfulness-based Psychoeducation [44]; Destigmatized Group Intervention [45]; CBT-based Psychoeducation Program [47]; Solution-focused Group Psychoeducation Program [49]; and Against Stigma Program [46]. These studies compared the intervention to Treatment as Usual or the Waiting List. Another study found substantial results when comparing Group Music Therapy to the Waiting List, but found no differences when compared to the Active Control condition of Group Education [42].

In terms of the effect size, 21 studies contributed their findings. However, four of them did not elaborate on the type of effect size they employed. Small effect sizes were found in three studies [53,55,59]; small to medium effect sizes were found in four studies [33,43,54,56]; medium effect sizes were found in five studies [30,34,48,52,60]; medium to large effect sizes were found in three studies [41,42,45]; and large effect sizes were found in two studies [32,62].

As for specific principal targets, Group Psychoeducation showed significant improvements in Perception [29], Knowledge about illness [31], and Insight [32], whereas NECT showed significant improvements in Social Withdrawal [48,54], Stereotype Endorsement, Alienation [48], Stereotype Agreement, Awareness, Application, and Self-esteem [34]. ESS showed significant improvements in Stigma Resistance, Recovery, Alienation, Stereotype Agreement, and Self-concurrence [51], while the Self-stigma Reduction Program showed significant improvements in Self-esteem, Readiness of Change, Participation [57], Perception, Self-efficacy, and Compliance [62].

Other notable interventions were Solution-focused Group Psychoeducation Program which showed significant improvements in Stereotype Endorsement, Stigma Resistance, Perceived Discrimination, Social Withdrawal, and Alienation [49]; Mindfulness-based Psychoeducation which showed significant improvements in Stereotype Endorsement, Perceived Discrimination, Social Withdrawal, and Alienation [44]; Coping Internalized Stigma Program (PAREI) which showed significant improvements in Recovery, Alienation, and Social Functioning [60]; Destigmatized Group Intervention which showed significant improvements in Perceived Discrimination, Social Withdrawal, and Alienation [45]; and CBT-based psychoeducation program which showed significant improvements in Stereotype Endorsement, Perceived Discrimination, and Alienation [47]. There were also other interventions that showed improvements in fewer principal targets [29,42,43,50,52,53,55,59].

### 3.2. Meta-Analysis

As this review is about determining the effectiveness of interventions for internalized stigma, a meta-analysis was conducted so that the significance of the link between the interventions that were utilized and their effectiveness in decreasing internalized stigma could be further investigated. Because the purpose of this review is not to ascertain which intervention is superior to the other, but rather to determine whether or not a particular intervention is effective when it is employed, studies that compared interventions to Treatment as Usual or a Waiting List were used. A total of 21 studies fit this description; however, only 18 of them were included in the analysis since 2 studies did not provide adequate extractable data. See Figure 2 for details.

The overall effect was statistically significant (Z = 3.00; *p* = 0.003; 95% CI: −0.69 [−1.15, −0.24]; n = 1633), although there was considerable heterogeneity (Tau^2^ = 0.89; Chi^2^ = 303.62, df = 17; *p* < 0.00001; I^2^ = 94%). This indicates that on the whole, the interventions that were utilized were effective in lowering internalized stigma in comparison to Treatment as Usual or the Waiting List; nevertheless, there was a significant amount of variation across these studies [40]. In view of the significant level of heterogeneity that exists across the studies, we moved on to the subgroup analysis [63,64].

#### 3.2.1. Subgroup Analysis

Due to the high I^2^, we conducted subgroup analyses in studies utilizing the same type of interventions.

##### Narrative Enhancement and Cognitive Therapy (NECT)

There were three studies that evaluated the effectiveness of Narrative Enhancement and Cognitive Therapy (NECT) to that of Treatment as Usual (TAU) or the Waiting List (WL) [33,34,48]. See Figure 3 for details. The summary effect was statistically significant (Z = 3.40; *p* = 0.0007; 95% CI: −0.44 [−0.70, −0.19]; n = 241), as well as it was highly homogenous (Tau^2^ = 0.00; Chi^2^ = 0.14, df = 2 (*p* = 0.93); I^2^ = 0%).

##### Group Psychoeducation

Four studies compared the effectiveness of Group Psychoeducation to that of Treatment as Usual (TAU) or the Waiting List (WL) [30,31,32,41]. See Figure 4 for details. The summary effect was not statistically significant (Z = 1.55; *p* = 0.12; 95% CI: −0.51 [−1.15, 0.13]; n = 297), as well as there was considerable heterogeneity (Tau^2^ = 0.36; Chi^2^ = 20.22, df = 3 (*p* = 0.0002); I^2^ = 85%). Even after further segmentation according to age, gender, duration of illness, and level of education, there was still high levels of heterogeneity.

### 3.3. Publication Bias

There was no indication of publication bias in any of the papers that we chose to include in the meta-analysis, as can be seen in the funnel plot that is shown in Figure 5.

### 3.4. NHRMC Evidence Statement Matrix

This review found that certain interventions may be effective methods for reducing internalized stigma; however, variations in these interventions needed to be considered as well. The overall rating of B for the NHRMC Evidence Statement Matrix was given. This indicates that there is some evidence in support of the recommendations, but caution should be exercised while implementing them. Appendix A provides a summary of the evidence matrix.

## 4. Discussion

This study explores a number of various psychological interventions that have the potential to reduce internalized stigma in individuals who have been diagnosed with schizophrenia spectrum disorders. To the best of our knowledge, this is the largest review to date on psychological therapies for internalized stigma in schizophrenia spectrum disorders, encompassing a total of 27 studies with a combined sample size of 2975 individuals from 15 different countries. Overall, the findings suggest that a variety of psychological therapies may be effective in reducing internalized stigma in people with schizophrenia spectrum disorders. A systematic review of studies revealed that 15 out of 27 trials yielded statistically significant results in favor of psychological therapies for decreasing internalized stigma. This is further supported by the meta-analysis of 18 trials, which demonstrated a significant effect of the therapies in reducing internalized stigma compared to Treatment as Usual or the Waiting List.

According to the findings, not only was Narrative Enhancement and Cognitive Therapy (NECT) somewhat effective from the standpoint of the systematic review, but it was also efficacious from the perspective of the meta-analysis that we conducted. These results are consistent with those of previous research that came to the conclusion that NECT is beneficial in reducing self-stigma as well as other subjective components of recovery such as hope and self-esteem in people who have suffered from psychotic-related illnesses [34,48,54]. NECT is a structured, group-based intervention that combines narrative therapy focused on enhancing one’s ability to narrate one’s life story, psychoeducation to help replace stigmatizing views about mental illness and recovery with empirical findings, and cognitive restructuring geared toward teaching skills to challenge negative beliefs about the self [34]. The length of time that the intervention was carried out ranged from five to six months. Since it has been seen phenomenologically that people with severe mental illness typically have a diminished capacity to narrate the unfolding tale of their own lives, NECT’s primary focus is on supporting the change in personal narratives. The ability to alter one’s narrative is seen as crucial in this context for the purpose of altering one’s sense of self. By having participants write or dictate tales about themselves and then receiving comments from the facilitator and group members on alternate viewpoints about the topics contained in their stories, NECT aims to assist individuals in reshaping their narratives. This intervention is effective because it is sensitive to the particular patient’s experiences and beliefs, and it also provides multiple tailored approaches to diminishing the internalized stigma in them.

Based on the findings of previous studies, Group Psychoeducation was the most commonly used psychological intervention for reducing internalized stigma [25,26,28]. These results coincide with the results of our study since we found that this intervention had been looked into in a total of five different studies [29,30,31,32,41]. In general, this intervention’s goal is to educate participants on a variety of important topics, such as gaining a better understanding of their condition, avoiding future relapses, discussing their experiences with stigma, and learning coping techniques to help them overcome stigma. The treatment not only combines psychodynamic strategies for coping with emotional reactions to the disease and stigma, but it also encompasses cognitive strategies for addressing attitudes and ideas about the illness. Previous research has demonstrated that psychoeducation is an effective method for helping people with schizophrenia to obtain a deeper understanding of their condition and its implications [65]. There are a few key distinctions between the use of this intervention among the trials that we studied, most notably with regard to the duration and frequency with which it is implemented. The shortest was seven sessions in three weeks [32], while the longest was twelve sessions in three months [41]. Despite positive findings from the systematic review on the effectiveness of this intervention, we discovered from our meta-analysis that the summary effect was not significant and that the heterogeneity was large. One possible explanation for this discrepancy is that the intervention was performed differently in various studies, with varied length and methodology. Consequently, this finding should be interpreted with caution due to the requirement to account for the variations in intervention.

It was discovered that among the most effective interventions were those that included a few different therapies. Combinations across therapies such as psychoeducation, cognitive behavioral therapy, social skills training, mindfulness, problem-solving skills, communication skills, and support groups were common. Among the interventions that had such multi-faceted elements were Self-stigma Reduction Program [62], Antistigma Photovoice Program [52], Mindfulness-based Psychoeducation [44], Destigmatized Group Intervention [45], CBT-based Psychoeducation Program [47], Solution-focused Group Psychoeducation Program [49], and Against Stigma Program [46]. Because of the many dimensions of the stigma, it addresses and the variety of approaches it provides, it appears that such therapies are advantageous in helping patients overcome internalized stigma. However, due to the limited number of studies that have been conducted to investigate their efficacy, accepting these conclusions has to be approached with caution.

Another notable matter is that the majority of the available interventions are carried out in groups rather than individually or in a combination of the two. This trend may have arisen for a variety of reasons, including the fact that group therapy is commonly viewed as more efficient financially and time-wise; patients gain comfort in knowing they are not alone in their struggles [66]; individuals gain insight through reflection on the experiences of others; and therapeutic alliances are strengthened through interactions with peers and facilitators during group sessions [67]. However, it is crucial not to overlook the possible advantages of interventions that are carried out individually, which may include a more intensive therapeutic experience, an individualized approach, and the maintenance of confidentiality. In light of this, it is essential to keep in mind that the effectiveness of an intervention may only be obtained to the fullest extent possible when the treatment is specifically adapted to the experiences and requirements of the patient.

Furthermore, internalized stigma is a multifaceted phenomenon that is comprised of a variety of components. One of the most effective strategies for overcoming it is to first identify the relevant principal targets that are involved, followed by prescribing interventions that are designed to explicitly address those targets while also taking into account many other aspects of the patient’s capabilities and requirements. For instance, we may use the Internalized Stigma of Mental Illness (ISMI) Scale to determine which parts of internalized stigma are primarily impacted in a patient. This can be carried out by analyzing the patient’s responses to the various questions on the scale. If Stereotype Endorsement and Alienation are a problem, then interventions such as NECT [48], Solution-focused Group Psychoeducation Program [49], Mindfulness-based Psychoeducation [44], or CBT-based psychoeducation program [47] that are known to effectively improve these targets may be utilized. Such tailored care may result in the following outcomes: an improved patient experience as a result of treatment decisions that take into account patients’ care needs and preferences; an improved population’s health and quality of life as a result of tailored care being supported; and a reduction in the per capita cost of care as a result of a reduction in the overuse, underuse, and misuse of healthcare services [68]. Furthermore, prior to implementing the intervention, it is essential to have a discussion about the options that are suitable and available, as well as the pros and cons of each, and to apply shared decision making between the patient and the doctor, as this may have many positive outcomes.

### Strength and Limitations

Among the strengths of this study is that, to the best of our knowledge, this is the largest review to date on psychological interventions for internalized stigma in schizophrenia spectrum disorders. In addition, this is also the first meta-analysis to examine the effectiveness of Narrative Enhancement and Cognitive Therapy (NECT) in lowering internalized stigma in people with schizophrenia spectrum disorders. However, there were some limitations to our study. Firstly, we limited our search to only include studies that had been published in English. As a result, we are unable to include the findings of other research that was published in a variety of languages, which leaves open the potential for publication bias. Secondly, the studies that were included in our research at times employed varying definitions of internalized stigma to some extent and also assessed it using different questionnaires that each had their unique method of ascertaining it.

Recommendations for future research include conducting more clinical trials testing the efficacy of each psychological intervention. This would allow for the measurement of pooled effects for each intervention, so providing a clearer picture of each’s effectiveness. After that, head-to-head comparisons of successful interventions might be conducted, yielding the most effective intervention. In addition, we also recommend that future research focus on the patient-tailored strategy of prescribing therapies based on the detected elements of internalized stigma.

## 5. Conclusions

According to this systematic review and meta-analysis, the majority of the psychological interventions evaluated are successful in lowering levels of internalized stigma, the use of NECT shows promise, and interventions that mix several therapies may be more beneficial. The strength of our research is that it is the largest review to date on psychological interventions for internalized stigma in people with schizophrenia spectrum disorders, as well as the first meta-analysis to evaluate the effectiveness of Narrative Enhancement and Cognitive Therapy (NECT) in lowering internalized stigma in people with schizophrenia spectrum disorders.

Insights from this study have the potential to give mental health practitioners more effective alternatives for implementing psychological therapies to reduce internalized stigma. Moreover, we recommend that future research focus on the patient-tailored strategy of prescribing therapies based on the detected elements of internalized stigma. In addition, we urge that future studies concentrate on the patient-tailored method of prescribing therapies based on the detected elements of internalized stigma.

## Figures and Tables

**Figure 1 ijerph-20-05570-f001:**
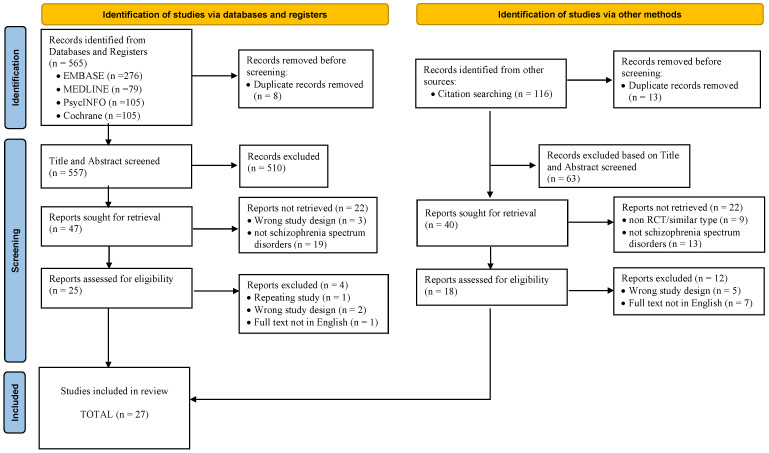
Flowchart based on the PRISMA 2020 statement [36] that describes the movement of information through the various stages of conducting this systematic review.

**Figure 2 ijerph-20-05570-f002:**
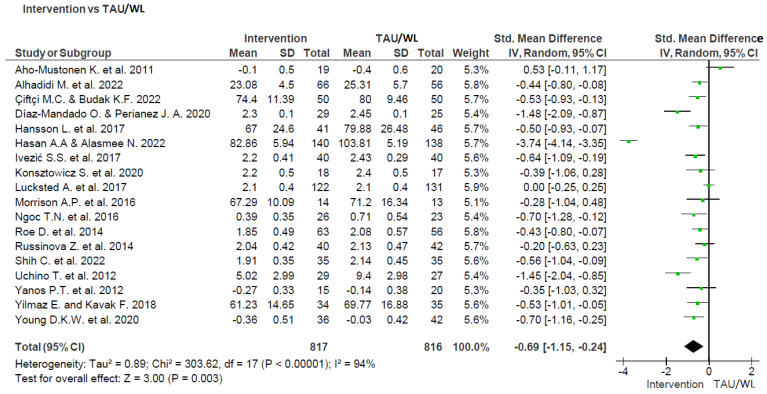
Meta-analysis and forest plot illustrating the efficacy of the overall interventions that were utilized in reducing internalized stigma in comparison to Treatment as Usual (TAU) or the Waiting List (WL) [30−34,41,43,45−48,51,52,55,58,59,61,64].

**Figure 3 ijerph-20-05570-f003:**
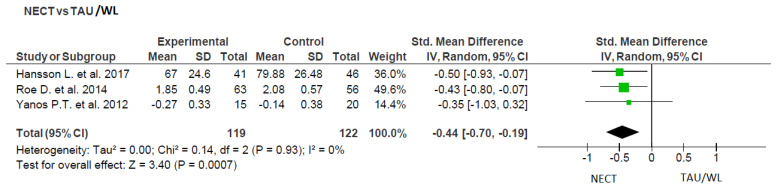
Subgroup analysis of Narrative Enhancement and Cognitive Therapy (NECT) in comparison to Treatment as Usual (TAU) or the Waiting List (WL) [33,34,48].

**Figure 4 ijerph-20-05570-f004:**
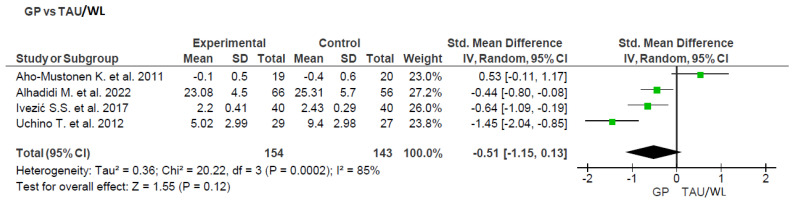
Subgroup analysis of Group Psychoeducation (GP) in comparison to Treatment as Usual (TAU) or the Waiting List (WL) [30,31,32,41].

**Figure 5 ijerph-20-05570-f005:**
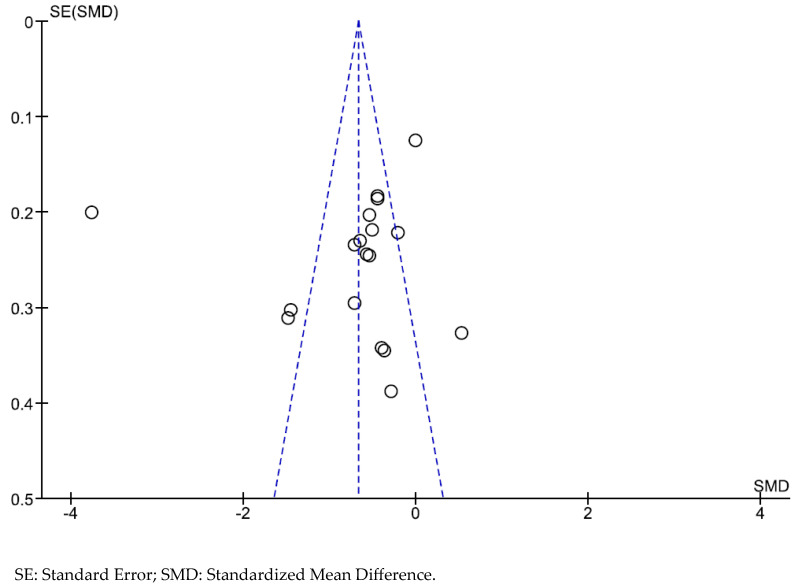
The funnel plot for meta-analysis on the overall interventions that were utilized in reducing internalized stigma in comparison to Treatment as Usual or the Waiting List.

**Table 1 ijerph-20-05570-t001:** Study Characteristics [29,30,32,33,34,41,42,43,44,45,46,47,48,49,50,51,52,53,54,55,56,57,58,59,60,61,62].

Study	Country	Study Design	Setting	Diagnosis	Sample Size	Intervention	Intervention N	Type	Frequency	Duration	Control	Control N	Type	Frequency	Duration	Outcome Measures	Main Result	Significance and Test used
**Shin S. & Lukens E.P. 2002** [29]	USA	RCT	outpatient	Schizophrenia	48	Group Psychoeducation	24	group	total 10 sessions (90 min/session)	10 weeks	Supportive Sessions	24	individual	total 10 sessions (45 min/session)	10 weeks	• SDS • BPRS • FCOPES	EG > CG	SoS = 2049.35; F = 376.10; df = 1,45; *p* < 0.001 ANCOVA
**Aho-Mustonen K. et al. 2011** [30]	Finland	RCT	inpatient	Schizophrenia, Schizoaffective	39	Group Psychoeducation	19	group	total 8 sessions (45–60 min/session)	8 weeks	TAU	20	NA	NA	NA	•PSQ • KASQ • SUMD • CRS • DAI-10 • BPRS • NOSIE-30 • BDI-II • RSE • 15D	CG > EG	*p* = 0.08 Cohen’s d = −0.59 Student’s *t*-tests
**Fung K.M.T 2011** [57]	China	RCT	outpatient	Schizophrenia	66	Self-Stigma Reduction Programme	34	group + individual	total 12group + 4 individual (2 group sessions/week)	NA	Newspaper reading	32	group + individual	total 12group + 4 individual (2 group sessions per week)	NA	• CSSMIS • PTCS • BPRS • GAF • CAQ-SPMI • SUMD • CGSS	EG > CG	F = 2.204; *p* = 0.096 Effect size = 0.096 Repeated measures ANOVA/ANCOVA with Bonferroni correction
**Uchino T. et al. 2012** [58]	Japan	NRCT	outpatient	Schizophrenia, Schizoaffective	56	Group Psychoeducation	29	group	total 6 sessions (1 session/week)	6 weeks	TAU	27	NA	NA	NA	• SDS-J • KIDI • DAI-10 • BPIS • GAF	EG > CG	*p* < 0.01 Mann-Whitney U-test
**Yanos P.T. et al. 2012** [33]	USA	RCT	outpatient	Schizophrenia, Schizoaffective, BMD, MDD	35	NECT	15	group	(1 h/session 1 session/week)	20 weeks	TAU	20	NA	NA	NA	• ISMI • BHS • RSES • CSC • QLS • PANSS • SUMD	EG = CG	F = 1.06; *p* = 0.312 Cohen’s d = 0.37 ANOVA
**Silverman M.J. et al. 2013** [42]	USA	pseudo-RCT	inpatient	MDD, BMD, Schizophrenia, Psychosis, Anxiety, PTSD	78	Music therapy (Mental Illness Stigma Educational Group Songwriting Session)	29	group	total 24 sessions (45 min/session ×1/week)	6 months	Group No-Music Educational Mental Illness Stigma Talk-based Session (Active Control Group) Wait-list Control Group (rock and roll bingo)	17 32	group group	total 8 sessions (45 min/session) total 8 sessions (45 min/session)	6 months 6 months	• SS	EG > WLCG EG = ACG	(df)F = (2,75)5.09; *p* = 0.008 ηp^2^ = 0.120 F test, Bonferroni inequality
**Roe D. et al. 2014** [48]	Israel	NRCT	NA	SMI	119	NECT	63	group	total 20 sessions (1 h/session ×1/week)	6 months	TAU	56	NA	NA	NA	• ISMI • MANSA • ADHS • RSE	EG > CG	F = 7.81; *p* = 0.006 ηp^2^ = 0.06 ANOVA
**Russinova Z. et al. 2014** [52]	USA	RCT	NA	Schizophrenia spectrum disorder, BMD, MDD, Others	82	Antistigma Photovoice Program	40	group	total 10 sessions (90 min/session)	10 weeks	Waiting List	42	NA	NA	NA	•ISMI •ACS •ES •GPSES •PGRS •CES-D	EG > CG	*p* = 0.03 Cohen’s d = 0.55 mixed-effects linear regression
**Morrison A.P. et al. 2016** [59]	England	RCT	outpatient	Schizophrenia spectrum disorder, Schizoaffective disorder, BMD with psychotic features, FEP, EIS	27	Cognitive Therapy	14	individual	total 12 sessions (1 h/session)	over a 4 month period	TAU	13	NA	NA	NA	• ISMI-R • SIMS • SS • QPR • BDI-7 • BHS • SIAS • SERS-S • ISS	EG = CG	F = 0.052; *p* = 0.822 Cohen’s d = 0.09 ANCOVA
**Ngoc T.N. et al. 2016** [55]	Vietnam	RCT	inpatient	Schizophrenia	49	Family Schizophrenia Psychoeducation Program (FSPP)	26	individual	total 3 sessions (1.5 h/session)	1.5 week	TAU	23	NA	NA	NA	• STS • QLESQ • MCI • CS	EG > CG	F(1,45) = 6.67; *p* < 0.05 R^2^ = 0.13 ANCOVA
**Hansson L. et al. 2017** [34]	Sweden	RCT	mixed	SMI	87	NECT	41	group	total 20 session (1 h/session)	NA	Waiting List	46	NA	NA	NA	• SSMIS-SF • RSES • MANSA	EG > CG	95% CI = 0.04–0.89; *p* = 0.013 Cohen’s d = 0.5 ANOVA
**Ivezić S.S. et al. 2017** [41]	Croatia	pseudo-RCT	outpatient	Schizophrenia	80	Group Psychoeducation	40	group	total 12 sessions	12 weeks	Waiting List	40	NA	NA	NA	• ISMI • BUES • PDD	EG > CG	F/t = 8.18; *p* = 0.005 ηp^2^ = 0.097 sample *t*-test
**Lucksted A. et al. 2017** [51]	USA	RCT	outpatient	Schizophrenia, BMD, Recurrent major depressive, Schizotypal, Borderline personality, Other delusional or psychotic disorder	253	Ending Self- Stigma Psychoeducational intervention	122	group	total 9 sessions (90 min/session)	9 weeks	TAU (minimally enhanced by a brochure about internalized stigma)	131	NA	NA	NA	• ISMI-29 • SSMIS • MARS • GSES • SOBI • BSI • ESS • BCI • R-BANS	EG > CG	df = 252; *p* = 0.131 Effect size = –0.139 repeated measures mixed model
**Wood L. et al. 2017** [53]	England	RCT	inpatient	Schizophrenia, Schizoaffective Schizophreniform, Delusional disorder, Psychotic disorder NOS, EIS	30	CBT	15	individual	total 1–2 sessions (2 h/session)	2 weeks	Psychoeducation	15	individual	total 1–2 sessions (2 h/session)	2 weeks	• ISMI-S • SS • SERS • QPR • BDI-PC • AMHP	EG = CG	F = 0.306; *p* = 0.585 Cohen’s d = −0.238 ANCOVA
**Li J. et al. 2018** [50]	China	RCT	outpatient	Schizophrenia	327	• SASD • Psycho-education • SST • CBT	169	individual	total 8 sessions (2 h/session)	9 months	Face-to-face Interview	158	individual	NA	9 months	• ISMI • DISC-12 • GAF • SQLS • SES • BPRS • PANSS-N	CG > EG	OR(95%CI) = −0.04(−0.14 to 0.06); *p* = 0.440 ICC = 0.1235 Linear Mixed Models
**Yilmaz E. and Kavak F. 2018** [44]	Turkey	NRCT	outpatient	Schizophrenia	69	Mindfulness-Based Psychoeducation	34	group	total of 12 sessions 2 days/week	6 weeks	TAU	35	NA	NA	NA	• ISMI	EG > CG	*p* = 0.001 Mann–Whitney U = 98.000 Independent-samples *t* test
**Gaebel W. et al. 2019** [56]	Germany	RCT	mixed	Schizophrenia spectrum disorders, Depression	462	Psychoeducational Group Therapy + STEM + Booster session	227	group	total 12 sessions (8 + 3 + 1)	4 to 11 weeks + 6w later booster	psychoeducational group therapy + booster session	235	group	total 12 sessions (11 + 1)	4 to 11 weeks + 6w later booster	• ISMI • HAM-D • PANSS • CGI • GAF • KCS • WHO-QOL-BREF • BUES • RSES • SCL-27	EG = CG	*p* = 0.83 d = 0.3 mixed models analyses and *t* tests
**Yanos P.T. et al. 2019** [54]	USA	RCT	mixed	Schizophrenia, Schizoaffective	118	NECT	65	group	total 20 sessions	20 weeks	Supportive Group Therapy	52	group	total 20 session	20 weeks	• ISMI • RSES • BHS • QLS • MSIF • PANSS • CSC • STAND	EG > CG	F = 2.19; df = 3347; *p* = 0.09 Effect Size = 0.25 (small-to-moderate) mixed-effects modeling, GPower3
**Díaz-Mandado O. & Perianez J. A. 2020** [60]	Spain	RCT	outpatient	Schizophrenia, Schizotypal, Persistent delusional disorder, Schizoaffective, Other non organic psychotic disorders, BMD, Recurrent depressive disorder, OCD	54	“Coping Internalized Stigma Program” (PAREI)	29	group	total 8 sessions (2 h/session ×1/week)	8 weeks	TAU	25	NA	NA	NA	• ISMI • PL • GI • RAS-41 • BPRS • SFS	EG > CG	F = 2.88; *p* = 0.1 ηp^2^= 0.06 ANOVA
**Drapalski A.L. et al. 2020** [61]	USA	RCT	outpatient	BMD, Schizophrenia, Schizoaffective, MDD with psychotic features	216	Ending Self-Stigma	106	group	total 9 session (75–90 min/session ×1/week)	9 weeks	Health & Wellness Intervention	110	group	total 9 session (75–90 min/session ×1 session/week)	9 weeks	• ISMI • SOBI • GSES • MARS • SSMIS	EG = CG	t = 0.72; *p* = 0.470 repeated-measures, mixed effects model
**Konsztowicz S. et al. 2020** [43]	Canada	pseudo-RCT	mixed	Schizophrenia, Schizoaffective, Other psychotic disorder, MDD with psychotic features, BMD with psychotic features, Unspecified schizophrenia spectrum and other psychotic disorder	35	Self-concept and Engagement in LiFe (SELF)	18	individual	50 min/session ×1/week	average of 4.6 weeks	TAU	17	NA	NA	NA	• ISMI • MES • RSQ • SERS • CDS • Q-LES-Q-18 • STQ • CSQ	EG > CG	F = 1.23; df = (1, 31); *p* = 0.28 ηp^2^ = 0.04 ANCOVA
**Young D.K.W. et al. 2020** [45]	China	NRCT	outpatient	Depression, Schizophrenia, BMD, Anxiety, Others	78	Destigmatized Group Intervention	36	group	total 10 sessions (90 min/session ×1 session/week)	10 weeks	TAU	42	NA	NA	NA	• ISMI • RAS • BDI	EG > CG	t(p) = −0.812(0.417); F(p) = 7.482(0.008) ηp^2^= 0.099 ANCOVA
**Alhadidi M. et al. 2022** [32]	Jordan	RCT	inpatient	Schizophrenia	122	Group Psychoeducation	66	group	total 7 sessions (45–60 min/session)	3 weeks	TAU	56	NA	NA	NA	• ISMI-10 • KASQ • BIS	EG > CG	F = 13.29, *p* < 0.001 d = –0.8 Repeat Measure ANCOVA
**Çiftçi M.C. & Budak K.F. 2022** [47]	Turkey	NRCT	outpatient	Schizophrenia	100	CBT-based Psychoeducation Program	50	group	total of 8 sessions (45–60 min/session ×2/week)	4 weeks	TAU	50	NA	NA	NA	• ISMI • FROGS	EG > CG	t = 2.672; *p* = 0.009 Effect Size = 0.8 power analysis, *t* test
**Erdoğan E. & Demir S. 2022** [49]	Turkey	RCT	outpatient	Schizophrenia	39	Solution-Focused Group Psychoeducation Program	20	group	total 6 sessions (90 min/session ×2/week)	6 weeks	TAU	19	NA	NA	NA	• ISMI • RSES • SubRAS	EG > CG	*p* < 0.001 W = 0.810 Spearman’s correlation analysis, Kendall W coefficient (effect size)
**Hasan A.A & Alasmee N. 2022** [62]	Jordan	RCT	outpatient	Schizophrenia spectrum disorder	278	Self-Stigma Reduction Programme	140	individual	total 13 sessions (biweekly)	26 weeks	TAU	138	NA	NA	NA	• ISMI • PANSS • PTCS • CGSS	EG > CG	F = 187.75; *p* < 0.001 Effect Size = 0.59 (large) ANOVA
**Shih C. et al. 2022** [46]	Taiwan	NRCT	inpatient	Schizophrenia	70	Against Stigma Program	35	group	60 min/session ×1/week	6 weeks	TAU	35	NA	NA	NA	• ISMI • RES	EG > CG	*p* = 0.012 Effect Size = 0.760 Generalized estimating equation

RCT: randomized controlled trial; NRCT: non-randomized, experimental trial; NA: not available; BMD: Bipolar Mood Disorder; OCD: Obsessive–Compulsive Disorder; MDD: Major Depressive Disorder; SMI: Severe Mental Illness (majority had psychotic disorder); FEP: First Episode Psychosis; EIS: Early Intervention Service; NOS: not otherwise specified; STEM: a psycho-educational group therapy module to promote stigma coping and empowerment; NECT: Narrative Enhancement and Cognitive Therapy; TAU: Treatment As Usual; EG: Experimental Group; CG: Control Group; WLCG: Wait-list Control Group; ACG: WLCG (Active Control Group); SDS:Link’s Stigma-Devaluation Scale; BPRS: Brief Psychiatric Rating Scale; FCOPES: Family Crisis Oriented Personal Evaluation Scales; PSQ: Perceived Stigma Questionnaire; KASQ: Knowledge about Schizophrenia Questionnaire; SUMD: Scale to Assess Unawareness of Mental Disorder; CRS: Compliance Rating Scale; DAI-10: Drug Attitude Inventory-1; NOSIE-30: Nurses’ Observation Scale for Inpatient Evaluation; BDI-II: Beck Depression Inventory-II; RSES: Rosenberg Self-Esteem Scale; 15D: 15D instrument of health-related quality of life; CSSMIS: Chinese Self-stigma of Mental Illness Scale; CAQ-SPMI: Change Assessment Questionnaire for People with Severe and Persistent Mental Illness; PTCS: Psychosocial Treatment Compliance Scale; GAF: Global Assessment of Functioning Scale; CGSS: Chinese General Self-efficacy Scale; SDS-J: Japanese version of the Social Distance Scale; KIDI: Knowledge of Illness and Drugs Inventory; BPIS: Birchwood’s Psychosis Insight Scale; ISMI: Internalized Stigma of Mental Illness Scale; BHS: Beck Hopelessness Scale; CSC: Coping with Symptoms Checklist; QLS: Quality of Life Scale; PANSS: Positive and Negative Syndrome Scale; MANSA: Manchester Short Assessment of Quality of Life; ADHS: Adult Dispositional Hope Scale; ACS: Approaches to Coping with Stigma scale; CES-D: Center for Epidemiological Studies Depression Scale; ES: Empowerment Scale; GPSES: Generalized Perceived Self-Efficacy Scale; PGRS: Personal Growth and Recovery Scale; ISMI-R: Internalized Stigma of Mental Illness Scale—Revised; SIMS: Semi-Structured Interview Measure of Stigma; QPR: Process of Recovery Questionnaire—Short Form; BDI-7: Beck Depression Inventory 7; SIAS: Social Interaction Anxiety Scale; SERS-S: Self-Esteem Rating Scale—Short Form; ISS: Internalized Shame Scale; QLESQ: Quality of Life Enjoyment and Satisfaction Questionnaire; STS: Stigma Towards Schizophrenia scale; MCI: Medication Compliance Inventory; CS: Consumer Satisfaction Scale; SSMIS-SF: Self-Stigma of Mental Illness Scale—Short Form; BUES: Boston University Empowerment Scale; PDD: Perceived Devaluation and Discrimination Scale; MARS: Maryland Assessment of Recovery in People with Serious Mental Illness; GSES: General Self-Efficacy Scale; SOFBI: Sense of Belonging Instrument; BSI: Brief Symptom Inventory; ESS: Experiences of Stigma Survey; BCI: Beck Cognitive Insight; R-BANS: Repeatable Battery of Neuropsychological Status; ISMI-S: Internalized Stigma of Mental Illness Inventory–Shortened; SS: Stigma Scale; SERS: Self-Esteem Rating Scale; BDI-PC: Beck Depression Inventory–Primary Care; AMHP: Attitudes towards Mental Health Problems; DISC-12: Discrimination and Stigma Scale; SQLS: Schizophrenia Quality of Life Scale; SES: Self-Esteem Scale; HAM-D: Hamilton Depression Rating Scale; CGI: clinical global status; KCS: Kemp Compliance Scale; WHO-QOL-BREF: World Health Organization Quality of Life Brief Version; SCL-27: symptom checklist; MSIF: Multidimensional Scale of Independent Functioning; STAND: Illness Awareness subscale of the Scale to Assess Narrative Development; PL: Perceived Legitimacy; GI: Group Identification; RAS: Recovery Assessment Scale; SFS: Social Functioning Scale; SSMIS: Self-Stigma of Mental Illness Scale; MES: Modified Engulfment Scale; RSQ: Recovery Style Questionnaire; CDS: Calgary Depression Scale; Q-LES-Q-18: Abbreviated Quality of Life Enjoyment and Satisfaction Questionnaire; STQ: Satisfaction with Therapy Questionnaire; CSQ: Client Satisfaction Questionnaire; BDI: Beck Depression Inventory; BIS: Birchwood Insight Scale; ISMI-10: Internalized Stigma of Mental Illness Inventory–10-Item Version; FROGS: Functional Remission of General Schizophrenia Scale; SubRAS: Subjective Recovery Assessment Scale.

**Table 2 ijerph-20-05570-t002:** Summary of the Findings of Interventions for Internalized Stigma [29,30,32,33,34,41,42,43,44,45,46,47,48,49,50,51,52,53,54,55,56,57,58,59,60,61,62].

Study	Quality of Study		Experimental Group	Control Group	Main Result	Internalized stigma	Stereotype Endorsement	Stigma Resistance	Insight	Empowerment	Recovery	Self-esteem	Anticipated Discrimination	Perceived Discrimination	Social Withdrawal	Readiness of Change	Participation	Stereotype Agreement	Self-concurrence	Awareness	Application	Disclosure	Knowledge about illness	Alienation	Internalised Shame	Hope	Engulfment	Social Functioning	Perception	Self-eficacy	Compliance
Shin S. & Lukens E.P. 2002 [29]	II A randomised controlled trial	60%	GP	SS	EG > CG	↑***																							✓***		
Aho-Mustonen K. et al., 2011 [30]	II A randomised controlled trial	87%	GP	TAU	CG > EG	↔			✓↔			✓↔											✓↔								
Fung K.M.T., 2011 [57]	II A randomised controlled trial	87%	SRP	NR	EG > CG	↔						✓*				✓*	✓*	✓↔	✓↔												
Uchino T. et al., 2012 [58]	III-2 A comparative study with concurrent controls: ▪ Non-randomised, experimental trial	80%	GP	TAU	EG > CG	↑**																	✓**								
Yanos P.T. et al., 2012 [33]	II A randomised controlled trial	100%	NECT	TAU	EG = CG	↔	✓↔	✓↔	✓↔																						
Silverman M.J. et al., 2013 [42]	III-1 A pseudorandomised controlled trial	87%	GMT	GE, WL	EG > WLCG EG = ACG	↑**								✓*								✓*									
Roe D. et al., 2014 [48]	III-2 A comparative study with concurrent controls: ▪ Non-randomised, experimental trial	80%	NECT	TAU	EG > CG	↑**	✓**								✓*									✓*							
Russinova Z. et al., 2014 [52]	II A randomised controlled trial	93%	APP	WL	EG > CG	↑*	✓*	✓**																							
Morrison A.P. et al., 2016 [59]	II A randomised controlled trial	87%	CT	TAU	EG = CG	↔					✓**														✓*	✓*					
Ngoc T.N. et al. 2016 [55]	II A randomised controlled trial	53%	FSPP	TAU	EG > CG	↑*																									✓**
Hansson L. et al., 2017 [34]	II A randomised controlled trial	87%	NECT	WL	EG > CG	↑*						✓**						✓*		✓*	✓*										
Ivezić S.S. et al., 2017 [41]	III-1 A pseudorandomised controlled trial	93%	GP	WL	EG > CG	↑**				✓↔																					
Lucksted A. et al., 2017 [51]	II A randomised controlled trial	93%	ESS	TAU	EG > CG	↔		✓*			✓*							✓**	✓**					✓*							
Wood L. et al., 2017 [53]	II A randomised controlled trial	93%	CBT	PE	EG = CG	↔					✓*	✓*																			
Li J. et al., 2018 [50]	II A randomised controlled trial	93%	SASD PE SST CBT	FTF	CG > EG	↔							✓*																		
Yilmaz E. and Kavak F. 2018 [44]	III-2 A comparative study with concurrent controls: ▪ Non-randomised, experimental trial	87%	MBP	TAU	EG > CG	↑***	✓***							✓***	✓***									✓***							
Gaebel W. et al., 2019 [56]	II A randomised controlled trial	87%	STEM	GP	EG = CG	↔				✓↔		✓↔																			
Yanos P.T. et al., 2019 [54]	II A randomised controlled trial	93%	NECT	SGT	EG > CG	↔	✓↔								✓*																
Díaz-Mandado O. & Perianez J. A. 2020 [60]	II A randomised controlled trial	87%	PAREI	TAU	EG > CG	↔	✓↔	✓↔			✓*			✓↔										✓**				✓*			
Drapalski A.L. et al. 2020 [61]	II A randomised controlled trial	87%	ESS	HW	EG = CG	↔						✓↔						✓↔			✓↔										
Konsztowicz S. et al. 2020 [43]	III-1 A pseudorandomised controlled trial	87%	SELF	TAU	EG > CG	↔					✓↔	✓↔															✓*				
Young D.K.W. et al. 2020 [45]	III-2 A comparative study with concurrent controls: ▪ Non-randomised, experimental trial	87%	DGI	TAU	EG > CG	↑**	✓↔							✓*	✓**									✓**							
Alhadidi M. et al. 2022 [32]	II A randomised controlled trial	93%	GP	TAU	EG > CG	↑***			✓***														✓NA								
Çiftçi M.C. & Budak K.F. 2022 [47]	III-2 A comparative study with concurrent controls: ▪ Non-randomised, experimental trial	87%	CBPP	TAU	EG > CG	↑**	✓**							✓*	✓↔									✓**							
Erdoğan E. & Demir S. 2022 [49]	II A randomised controlled trial	100%	SFGPP	TAU	EG > CG	↑***	✓***	✓***						✓**	✓***									✓***							
Hasan A.A & Alasmee N. 2022 [62]	II A randomised controlled trial	87%	SRP	TAU	EG > CG	↑***																							✓*	✓***	✓***
Shih C. et al. 2022 [46]	III-2 A comparative study with concurrent controls: ▪ Non-randomised, experimental trial	87%	ASP	TAU	EG > CG	↑*						✓***																			

↑ = statistically reduces internalized stigma; * = *p* < 0.05; ** = *p* ≤ 0.01; *** = *p* ≤ 0.001; ↔ = statistically no difference; ✓ = noted improvement in principal; GP = Group Psychoeducation; SS = Supportive Sessions; TAU = Treatment As Usual; SRP = Self-stigma Reduction Program; NR = Newspaper Reading; NECT = Narrative Enhancement and Cognitive Therapy; GMT = Group Music Therapy; GE = Group Education; WL = Waiting List; APP = Antistigma Photovoice Program; CT = Cognitive Therapy; FSPP = Family Schizophrenia Psychoeducation Program; ESS = Ending Self-Stigma psychoeducational intervention; CBT = Cognitive Behavioral Therapy; PE = Psychoeducation; SASD = Strategies Against Stigma and Discrimination; SST = Social Skills Training; FTF = Face-to-face interview; MBP = Mindfulness-based Psychoeducation; STEM = a psycho-educational group therapy module to promote stigma coping and empowerment; SGT = Supportive Group Therapy; SELF = Self-concept and Engagement in LiFe; PAREI = Coping Internalized Stigma Program; DGI = Destigmatized Group Intervention; HW = Health and Wellness Intervention; SFGPP = Solution-focused Group Psychoeducation Program; ASP = Against Stigma Program; CBPP = CBT-based Psychoeducation Program; EG = Experimental Group; CG = Control Group; WLCG = Wait-list Control Group; ACG = Active Control Group.

## Data Availability

Not applicable.

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
