# Peer review of "Evidence for the Effectiveness of Psychological Interventions for Internalized Stigma among Adults with Schizophrenia Spectrum Disorders: A Systematic Review and Meta-Analyses"

_ijerph, 2023, doi:10.3390/ijerph20085570_

Round 1
Reviewer 1 Report
The manuscript reports findings of a systematic review and meta-analysis on the efficacy of the psychological interventions targeting internalized stigma among adults with schizophrenia spectrum disorders. The work demonstrated the efficacy of psychological interventions in the reduction of internalized stigma, which is highly prevalent among individuals with severe mental disorders like schizophrenia. The work was meticulously conducted, and the findings were clearly presented. The findings are clinically relevant, bringing implications for the recovery and remission of individuals with schizophrenia spectrum disorders.
I have some comments, mainly on the Methods and Results of the manuscript, for the authors’ consideration, as follows:
Introduction:
- An overview of the psychological treatments to reduce stigma and their underlying mechanisms of change, probably based on the conceptualization of the internalized stigma in the first paragraph, would be helpful to pin down the scope of the review, as well as the contribution and scientific value of the work. Somewhere in the second last paragraph of the Introduction will do.
Methods:
- In line 145 (p. 3), please clarify “similar type study design”. The phrase is too precise enough.
- Some details on how the McMaster Critical Appraisal Tool for Quantitative Studies assesses the level of quality should be included in Section 2.4.
- I wondered if the subgroup analysis by types of interventions was intentional or ad-hoc (ref sections 2.6 & 3.2.1). The registered protocol mentions that subgroup analyses “will be done by age (<20, 20-30, 30-40, >40), definitive diagnosis and types of internalized stigma (perceived stigma and self-stigma)”, which did not include the types of interventions. This seems to be a violation of the registered protocol, which needs some justifications.
Results:
- In Figure 1 (PRISMA diagram), the phrase “Studies included in review” and “Reports of included studies” in the bottom box seems redundant to me, although I am aware that in total 27 studies, from electronic databases and secondary search, were included in the review.
- Please state the exact number of studies when referring to the “most common type” (line 323) and “second most common type” (line 325) of interventions in Section 3.1.4.
- In lines 359 to 361, it is mentioned that “validity and reliability of the internalized stigma measures that have been utilized are still dubious, due to the fact that only a select handful have been subjected to rigorous testing.” Do you mean that some of the measures of internalized stigma have questionable psychometric properties? It would be important to make a case for which measure(s) should be included in the review, to avoid contamination of results due to the lack of psychometric soundness of the measures.
- In Section 3.2, the authors stated that “studies that compared interventions to Treatment as Usual or Waiting List were used”. This is a bold methodological decision, which should be stated and explained earlier in the Methods section. It is also important to let the readers know that the meta-analysis only covered a subset of included studies, while the systematic review had a wider scope.
- In lines 468-470 (under section 3.2), the procedure of meta-analysis of correlation was mentioned. It came as a sudden here, as it is not explained in the Method section. Please re-arrange the statement, or delete it if it is not relevant.
- The procedure for testing publication bias should also be mentioned in the Method section for the completeness of the description of the methodology. Also, please consider statistical tests of publication bias, such as Egger’s test and other variants.
Discussion:
- The statement “among the most effective interventions were those that included a few different therapies” (lines 576-577, p.6) needs some support from statistical analysis (maybe I might have missed that part of the results).
- The second limitation echoed my previous enquiry regarding the psychometric properties of (some of) the measures of internalized stigma. Indeed, it is not surprising that various measures may capture the concept differently. What is methodologically relevant here is to have a standard definition of “internalized stigma” in the review (which has been clearly explained in the first paragraph of the Introduction), and then check if these measures align with the definition, as an inclusion/ exclusion criterion of the studies.
Besides, I recommend that authors check the whole manuscript, especially the Introduction, regarding the use of words and clarity of the statements. Here are some cases of sample statements in which revision or clarification is needed.
- “Thus, perceived stigma is the first stage of self-stigma formation, according to this model; they are both involved in internalizing public stigma.” (lines 46-48). What does “they” refer to?
- “Schizophrenia is a renowned, serious mental illness that affects various groups in the community” (lines 52-53). What are these “various groups”?
- “The outcomes of interventions at different points in time or in different regions varied greatly. This might be because of numerous variables, including small sample sizes, variation in intervention implementation, anecdotal definitions of stigma, and so on” (lines 107-109). The meaning of the phrases “different points in time” and “numerous variables” are not clear.
Reviewer 2 Report
I suggest doing a discussion in differents epigraphes to compare with the differents results.
Reviewer 3 Report
This is a very interesting paper investigating the effect of psychotherapy on internalized stigma in adults diagnosed with schizophrenia and related disorders. The paper is well-written and of interest for the journal; however, several minor changes are recommended before considering it for publication.
ABSTRACT
1- The authors carried out a systematic review and meta-analysis. I recommend to include which databases were electronically searched in the abstract. They mention that they were 4.
2- Further analyses were done by using the RevMan software. I recommend to metnion "Further quantitative analyses".
INTRODUCTION
1- The authors start the review by introducing the concept of internalized stigma. Afterwards, they describe several aspects on schizophrenia. I recommend to begin with the concept of schizophrenia, and follow with the description of the WHO.
2- At the end of the introduction section, the authors describe the aim of the study. I recommend to expand this explanation, and build a new subsection for the Aims. Example: 1.1. Aims.
MATERIAL AND METHODS
1- It is not necessary to repeat the number of papers that were recorded, screened, etc, in the results section. Please, refer to the Figure 1.
DISCUSSION
1- In the discussion section (line 590) the authors describe " An additional conclusion that is notable is that...". I recommend not to use these words because in the discussion section there should not be described conclusions.
2- What are the authors recommending for future studies? Is there a combination of psychotherapies that can be recommended for these patients? Is internalized stigma necessary to be investigated simultaneously with neurocognitive performance?
CONCLUSIONS
The conclusions should be a summary of the paper.
Round 2
Reviewer 1 Report
As noted in my initial review, the manuscript reports a systematic review and meta-analysis of the efficacy of interventions targeting internalized stigma among individuals with schizophrenia spectrum disorders. I appreciate the authors’ effort on providing a thorough response to my and other reviewers' suggestions and a clear summary of the revisions. I have no further recommendations.